# Incremental Construction of Motorcycle Graphs

**Franz Aurenhammer, Christoph Ladurner and Michael Steinkogler ***

Institute for Theoretical Computer Science, Graz University of Technology, 8010 Graz, Austria;
auren@igi.tugraz.at (F.A.); christoph.ladurner@student.tugraz.at (C.L.)
**\*** Correspondence: michael_steinkogler@gmx.net

**Abstract:** We show that the so-called motorcycle graph of a planar polygon can be constructed by a randomized incremental algorithm that is simple and experimentally fast. Various test data are given, and a clustering method for speeding up the construction is proposed.

**Keywords:** planar polygon; motorcycle graph; straight skeleton; incremental algorithm; randomization





## 1. Introduction

The so-called *straight skeleton* is an internal tree structure for polygons. It is used in computational geometry and also in other fields of science [1] as a piecewise-linear alternative to the (potentially curved) medial axis of a polygon. The straight skeleton is defined by a mitered offsetting process, where the boundary of the polygon gets moved inwards in a self-parallel manner. Thereby, the polygon edges change in length and eventually shrink to length zero and disappear, and the shrinking polygon may split at various places. During the offsetting process, the vertices of the polygon move on internal angle bisectors and trace out certain straight-line segments in the polygon's interior that build up a tree structure—the straight skeleton.

The *reflex* vertices of the polygon (i.e., those having an internal angle larger than $\pi$) play a special role in the construction of the straight skeleton. The reason is that such vertices are responsible for possible polygon splits, which are non-local events and therefore are (unlike the disappearances of edges) costly to predict. The problem of dissolving the interaction of the reflex polygon vertices was abstracted by Eppstein et al. [2] into a separate problem which they called the *motorcycle graph* problem. Figures 1 and 2 give an example and explanations. Consider $r$ 'motorcycles' starting from $r$ given points in the plane (the reflex polygon vertices), at different but constant speeds (given by the interior polygon angles) and in different directions (following the respective angle bisectors). Each motorcycle produces a trace where other motorcycles, when running into it, crash and end their movement. The collection of all the motorcycle traces now define the motorcycle graph of the polygon.

Computing the straight skeleton or at least the motorcycle graph efficiently has been a topic in computational geometry over the years; see e.g., [2–5]. Indeed, precomputing the motorcycle graph lead to the first straight skeleton algorithm with sub-quadratic runtime. Typically, the behaviour of the motorcycles is tracked over time, handling all motorcycles simultaneously—the simplest algorithm just computing all the intersections between traces beforehand.

The goal of the present paper is to show that treating the motorcycles individually (rather than simultaneously) leads to a simple and practical algorithm. We will apply *randomized insertion* to the motorcycles and maintain a partial motorcycle graph during this process. Our empirical results indicate competitiveness of our new method to other implemented algorithms like in [6]. A conference version of this paper appeared in [7].

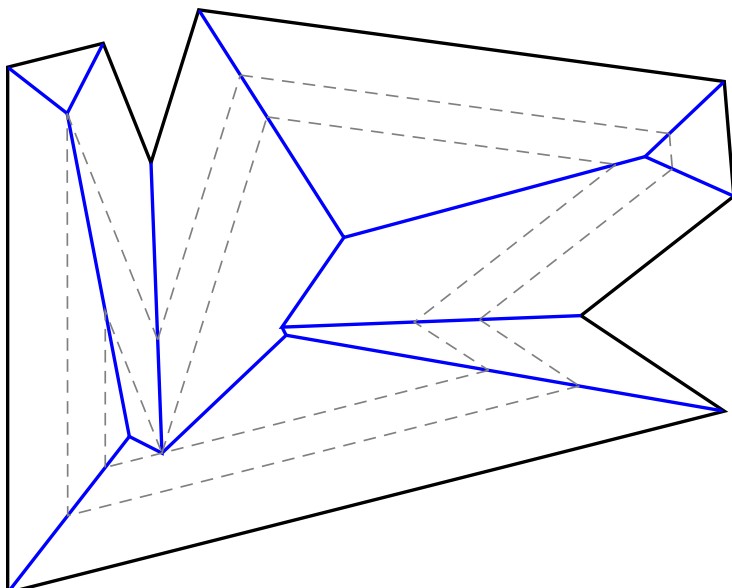

**Figure 1.** Straight skeleton of a polygon. The dashed polygon offsets indicate its shrinking process.

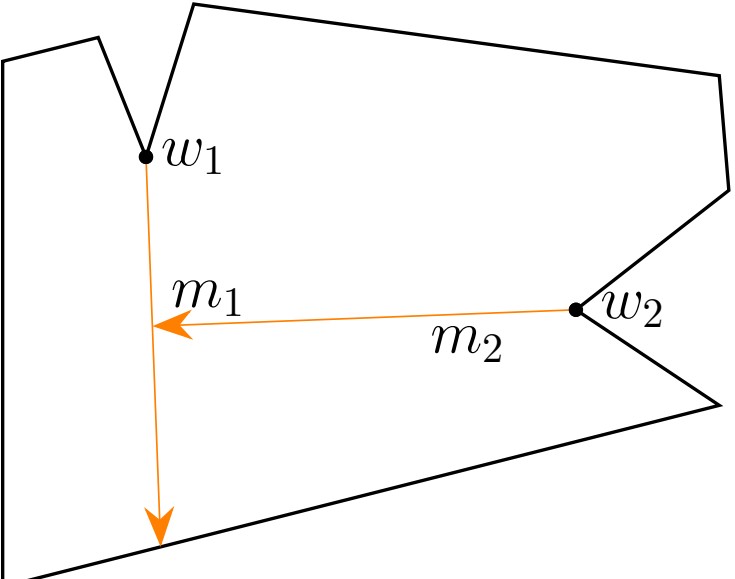

**Figure 2.** The corresponding motorcycle graph is defined by two motorcycles, $m_1$ and $m_2$, that start at the reflex vertices $w_1$ and $w_2$, respectively.

## 2. Insertion Algorithm

### 2.1. Preliminaries

In this work, $\mathcal{P}$ is always a simple polygon with $n$ vertices, $\partial\mathcal{P}$ denoting its boundary. Let $W = \{w_1, \dots, w_r\}$ be the subset of all the *reflex* vertices of $\mathcal{P}$, in some fixed order. With each such reflex vertex $w_i \in W$ we associate a constant velocity $\overrightarrow{v_i}$. (In the context of the straight skeleton of the polygon $\mathcal{P}$, this velocity is defined as the speed and direction of $w_i$ during the shrinking process of $\mathcal{P}$). Furthermore, let $m_i$ be the motorcycle moving from $w_i$ with velocity $\overrightarrow{v_i}$. Then at time $t > 0$, the motorcycle $m_i$ is located at $w_i + t \cdot \overrightarrow{v_i}$. We assume that $m_i$ leaves behind a straight trace, and when $m_i$ reaches the trace of another motorcycle or the polygon boundary $\partial\mathcal{P}$ at some time $t$, then $m_i$ is said to *crash*: That is, for all $t' > t$, the position of $m_i$ remains at $w_i + t \cdot \overrightarrow{v_i}$. The union of all motorcycle traces generated in this way is called the *motorcycle graph* of $W$, or $\mathcal{M}(W)$ for short.

### 2.2. Localized Intersection Computations

In [4] Cheng and Vigneron introduced a partition of the polygon $\mathcal{P}$ (or more generally, of the entire plane) into convex cells, in order to reduce the number of intersection computations between motorcycle traces. The positions of the motorcycles are tracked over time, and intersections are only computed among motorcycle traces that are 'active' within the same cell. A motorcycle is called *active* in a cell $C$ at time $t$ when it was located within $C$ for some time $t' \leq t$. The set of active motorcycles is tracked for each cell as a local arrangement of line segments $\mathcal{A}(C)$, initialized to the empty set at the start of the algorithm. If a motorcycle $m$ crashes at some time $t$, its 'death time' is recorded as $d(m) = t$; initially $d(m)$ is set to $\infty$, and $m$ is called *alive* if it has not crashed yet. To track the state of the motorcycles over time, three kinds of events are processed: *collision* events (two motorcycle traces intersect within a cell; these events are called impact events in [4]), *switch* events (a motorcycle crosses the cell boundary to enter the next cell), and *boundary* events (the motorcycle crashes into the polygon boundary; these events are not considered in [4] because the motorcycles are not restricted to a polygon there). The events are managed by a time-ordered queue $\mathcal{Q}$, initialized with switch events taking each motorcycle to its starting cell (i.e., the cell adjacent to the reflex polygon vertex where the motorcycle starts). The events are processed in the following way:

**Boundary** event of motorcycle $m$ at time $t$, where $m$ crashes into $\partial \mathcal{P}$ at some point $q$. If $m$ is alive at $t$, the motorcycle edge $\overline{wq}$ is reported. In addition, we put $d(m) = t$. If $m$ was already dead at time $t$ there is nothing to do.

**Collision** event of motorcycle $m$ at time $t$, where $m$ reaches the trace of another motorcycle $m'$ at point $q$, with $m'$ having tentatively passed $q$ at some time $t' < t$. If $d(m) < t$ ($m$ has already crashed) or if $d(m') < t'$ ($m'$ has crashed before reaching $q$), then there is nothing to do. Otherwise $m$ crashes at $q$, and so we set $d(m) = t$.

**Switch** event of motorcycle $m$ at time $t$, moving from cell $C$ to cell $C'$. If $d(m) < t$ there is nothing to do. Otherwise the line segment representing the (tentative) track of $m$ through $C'$ is inserted into the arrangement $\mathcal{A}(C')$. Collision events of $m$ with other motorcycles in $C'$ are computed via $\mathcal{A}(C')$ and are inserted into the queue $\mathcal{Q}$. (Note that collision events between two motorcycles are always associated with the motorcycle reaching the collision point last). Also, the next switch event for $m$, passing from $C'$ to the next cell $C''$, is computed and inserted into $\mathcal{Q}$. If there is no such cell $C''$, i.e., when $m$ crashes into $\partial \mathcal{P}$, then the corresponding boundary event is inserted into $\mathcal{Q}$.

The running time of this algorithm is largely dependent on the chosen cell partition. By using an $\frac{1}{\sqrt{n}}$ cutting, Cheng and Vigneron achieve a running time of $\mathcal{O}(n\sqrt{n}\log n)$. They also present a simpler randomized approach: choose randomly $\sqrt{n}$ motorcycles, and use the arrangement of the support lines defined by the chosen motorcycles as the partition. This results in an expected running time of $\mathcal{O}(n\sqrt{n}\log n)$. A slight variation of this idea has been used in an implementation by Huber and Held in [6], where they partition the plane using a $\sqrt{n}$ times $\sqrt{n}$ rectangular grid.

Common to these approaches is the *simulateneous* processing of all motorcycles over time. By contrast, we shall consider each motorcycle at a time: We devise an insertion algorithm for the motorcycles, which is based on a cell partition different from the ones used before, and which is described below.

### 2.3. Motorcycle Insertion

Let $W_k = \{w_1, \ldots, w_k\}$ be a subset of the reflex vertices of $\mathcal{P}$, and let $\mathcal{M}(W_k)$ be the *partial* motorcycle graph, where only the $k < r$ motorcycles of the vertices in $W_k$ move. In this section we show how the next motorcycle, $m_{k+1}$, can be inserted into $\mathcal{M}(W_k)$ in order to obtain $\mathcal{M}(W_{k+1})$.

In the simplest case, $m_{k+1}$ crashes at $\partial\mathcal{P}$ or at the first motorcycle trace in $\mathcal{M}(W_k)$ that it reaches. Then we only need to add the motorcycle edge of $m_{k+1}$ to $\mathcal{M}(W_k)$ to obtain $\mathcal{M}(W_{k+1})$. Otherwise, the insertion of $m_{k+1}$ causes more complex structural changes. In the worst case, all the edges of $\mathcal{M}(W_k)$ can change, meaning that they either get shortened or extended in $\mathcal{M}(W_{k+1})$.

To compute these changes, we adapt the algorithm from [4]. However, we use the (already computed) partial motorcycle graph $\mathcal{M}(W_k)$ as our cell partition of $\mathcal{P}$. Initially, there is only one moving motorcycle, $m_{k+1}$. $\mathcal{Q}$ is initialized with a single event, the switch event for $m_{k+1}$ when it enters the cell adjacent to $w_{k+1}$. As cell boundaries correspond to motorcycle traces, switch and collision events can happen at the same time; in such a case the collision event is processed first. The event processing is adapted in the following way:

**Boundary**  events are handled as before.

**Collision**  event of motorcycle $m$ at time $t$, reaching the trace of motorcycle $m'$ at point $q$, with $m'$ having tentatively passed $q$ at some time $t' < t$. This is handled as before, and in case both motorcycles are alive at $q$ (i.e., $m$ actually crashes at $q$) and $m \neq m_{k+1}$, then we additionally need to 'activate' all motorcycles $\widetilde{m}$ that got blocked by $m$ (after $m$ passed $q$) and that have $d(\widetilde{m}) > t$. See Figures 3 and 4 for illustrations. For all such motorcycles $\widetilde{m}$ we do the following: Put $d(\widetilde{m}) = \infty$, and insert into $\mathcal{Q}$ a switch event of $\widetilde{m}$ crossing the edge defined by $m$ to reach the next cell, say $C'$. (This switch event will then add $\widetilde{m}$ to $A(C')$.)

**Switch**  events are handled as before.

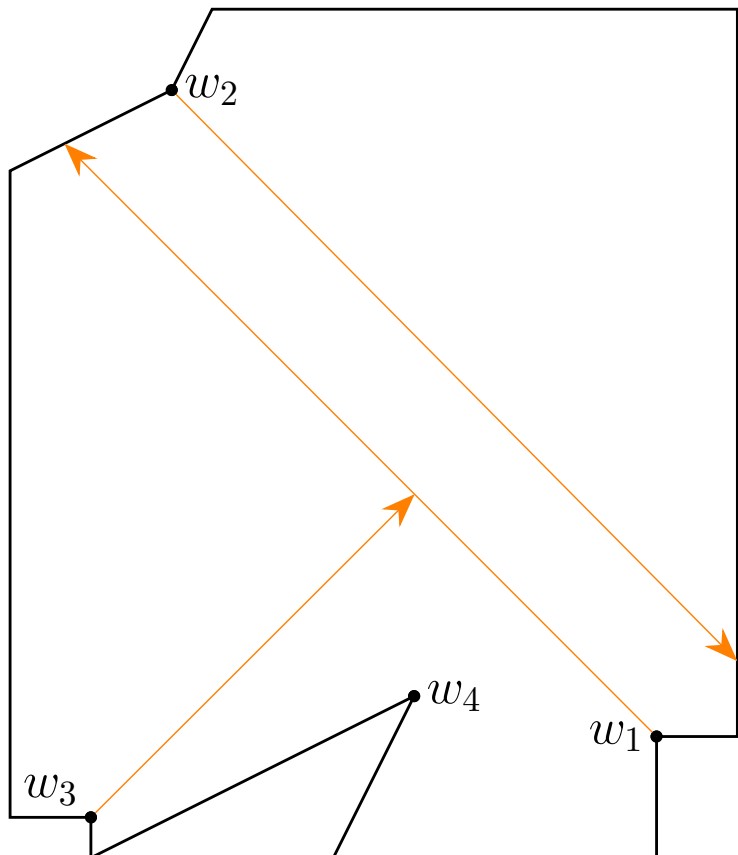

**Figure 3.** Partial motorcycle graph before the insertion of motorcycle $m_4$.

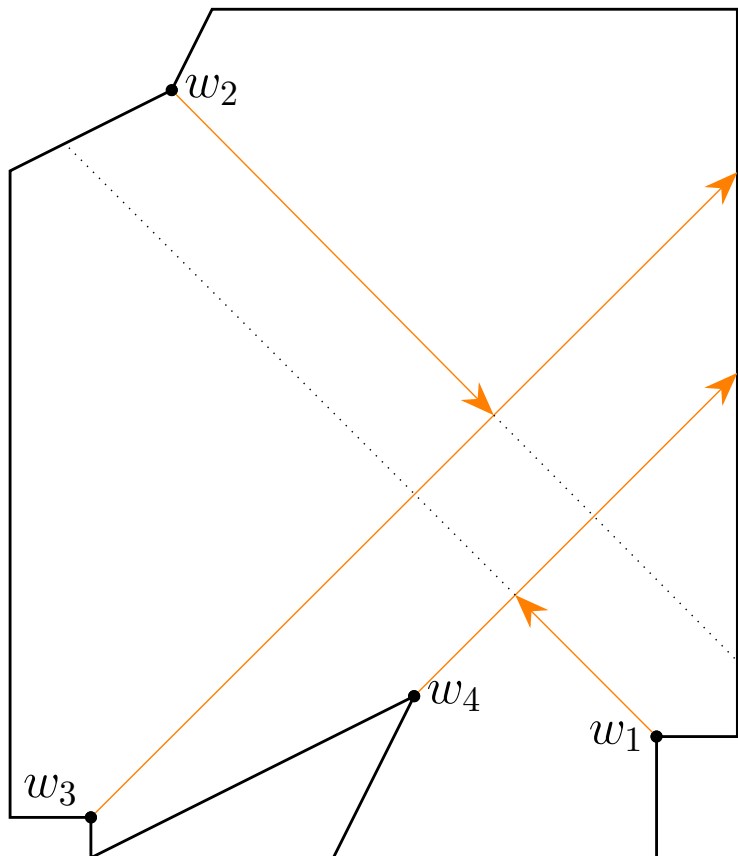

**Figure 4.** Insertion of $m_4$ makes $m_1$ crash. This activates $m_3$ which is now blocking $m_2$.

**Theorem 1.** *The motorcycle insertion algorithm is correct.*

**Proof.** Correctness of our approach mainly follows from the correctness of the Cheng-Vigneron algorithm [4]. The only difference is that our algorithm avoids creating and processing certain events because $\mathcal{M}(W_k)$ is used as the underlying polygon partition.

Consider all collision events for the motorcycle graph $\mathcal{M}(W_k)$, as well as all the switch events for the motorcycles $m_1, \ldots, m_{k+1}$. These events are precomputed in [4], as opposed to our on-demand approach. Our algorithm processes the same events, except that we need not process the collision events for $\mathcal{M}(W_k)$ because we already know the result of these events (namely, $\mathcal{M}(W_k)$ itself)—if these events are relevant at all (motorcycles may crash earlier in $\mathcal{M}(W_{k+1})$). Also, computing switch events on demand—when a motorcycle enters a new cell—for $m_{k+1}$ and for the unblocked motorcycles of $\mathcal{M}(W_k)$ changes neither the events nor their processing. Thus, our algorithm and the algorithm of [4] produce the same result. $\square$

Clearly, the final result of our algorithms does not depend on the order in which the motorcycles are inserted. While intermediate results (i.e., the partial motorcycle graphs $\mathcal{M}(W_k)$) might be different, the algorithm always ends up with constructing $\mathcal{M}(W)$. We will exploit this fact and will insert the motorcycles in *random order* to reduce the running time in the expected case.

### 2.4. Partition Maintenance

So far we discussed the insertion of a single motorcycle $m_{k+1}$ into $\mathcal{M}(W_k)$ to compute $\mathcal{M}(W_{k+1})$, but to obtain an algorithm for incrementally constructing the motorcycle graph we need to maintain the partial motorcycle graph during the insertion steps. From $\mathcal{M}(W_k)$ to $\mathcal{M}(W_{k+1})$ existing motorcycle edges can be shortened or extended. Shortening an edge causes two adjacent cells to be merged, extending an edge causes a cell to be split (see

Figure 5 for an illustration). During the event processing, the algorithm needs to compute switch and boundary events—this can be done by using ray shooting queries within cells. And when inserting a new motorcycle $m_{k+1}$ it is necessary to find the adjacent cell of $\mathcal{M}(W_k)$ for the initial switch event.

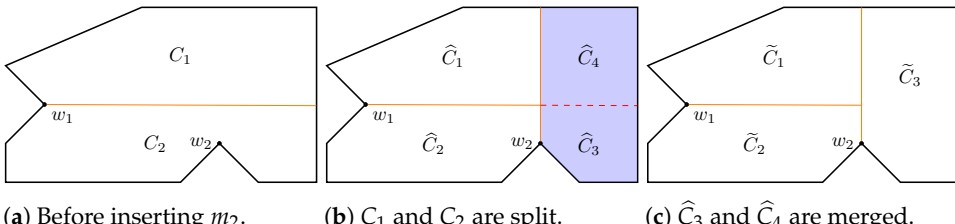

(**a**) Before inserting $m_2$.    (**b**) $C_1$ and $C_2$ are split.    (**c**) $\widehat{C}_3$ and $\widehat{C}_4$ are merged.

**Figure 5.** Updating the cell structure of the motorcycle graph.

The data structure from Goodrich and Tamassia [8] supports all needed operations in $\mathcal{O}(\log^2 n)$ time using $\mathcal{O}(n)$ space. In practice, if costly operations do not occur too often, it may be efficient enough to organize $\mathcal{M}(W_k)$ in a doubly-connected edge list data structure, and to do simple boundary scans for all operations (see the Section 3 for data supporting this approach). Also note that, while the cells of $\mathcal{M}(W_k)$ are in general not convex, they can be treated as such for in-cell ray shooting queries: Figure 6 explains how a cell can be extended to a convex cell for this purpose.

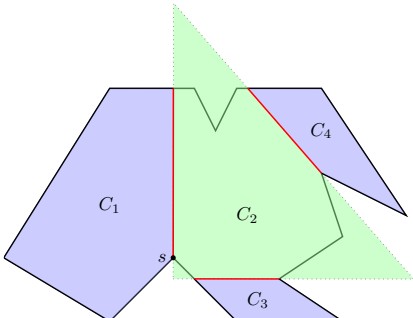

**Figure 6.** Motorcycle cells ($C_1$ to $C_4$) induce (possibly infinite) convex polygons by extending their bordering motorcycle edge segments. (The green area shows the induced polygon for the cell $C_2$.).

## 3. Experimental Results

Analysing our insertion step, we can observe that $\Theta(k)$ structural changes can happen from $\mathcal{M}(W_k)$ to $\mathcal{M}(W_{k+1})$. This basically amounts to a complete recomputation of the partial motorcycle graph. Figure 7 gives an example. Assuming this worst case for all iterations, we end up with a running time that is a factor of $n$ worse than the algorithm from [4].

The construction given in Figure 8 has as many as $\Theta(n)$ structural changes in the last insertion step, averaged over *all* reflex vertices. This shows that, even when using randomized insertion, the expected runtime of a single step can be as large as $\Theta(k)$. As a consequence, applying the backwards analysis technique for randomized incremental insertion (as in [9]) does not provide any improved runtime bounds.

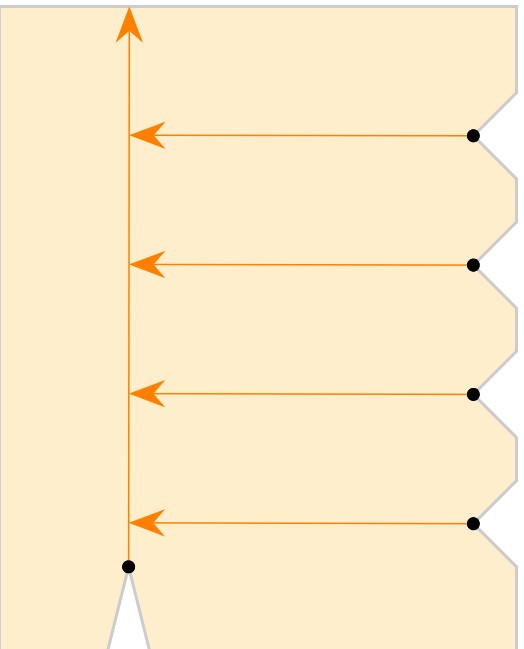

**Figure 7.** The insertion of one motorcycle can block all the others.

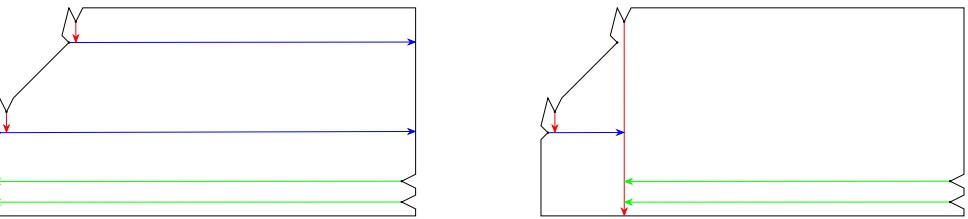

**Figure 8.** Polygon angles can be adjusted such that removal of any of the $\Theta(n)$ blue motorcycles unblocks its associated red motorcycle, which in turn blocks the $\Theta(n)$ green motorcycles.

Deriving theoretical bounds on the expected overall runtime thus seems to be intricate, and we instead explored the practical performance of our randomized insertion algorithm on large sets of polygons of different shapes and characteristics. We observed the occurring number of structural changes—for each already inserted motorcycle a *modification counter* for its associated edge in the partial motorcycle graph is maintained.

The polygons considered can be divided into three categories: Random polygons from the *Salzburg Database of Geometric Inputs* [10] (like in Figure 9), polygonal country outlines taken from Natural Earth [11] and DIVA-GIS [12] (see Figure 10 for examples), and specifically constructed polygons (see Figures 8 and 11). Many more details on the used data and their exploration can be found in Ladurner's Master thesis [13].

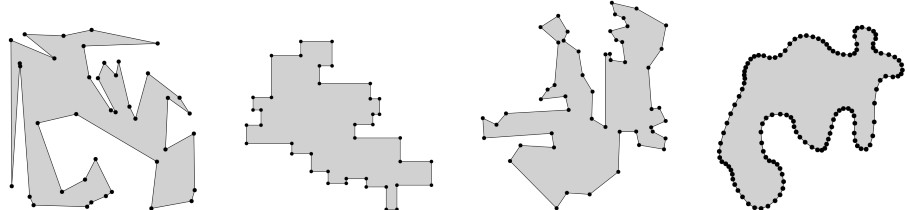

**Figure 9.** Examples of random polygons from *Salzburg Database of Geometric Inputs* [10].

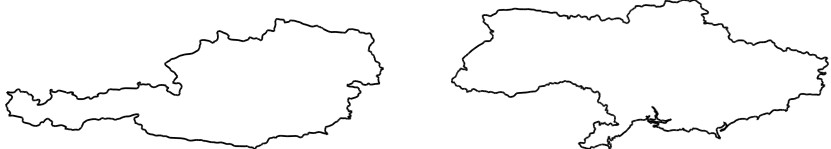

**Figure 10.** Examples of country-based polygons from *Natural Earth* [11].

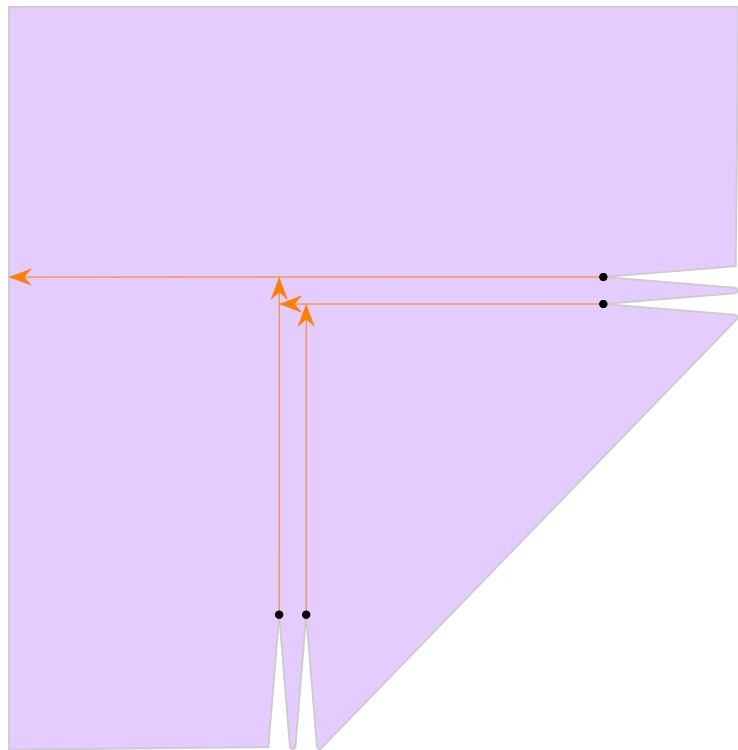

**Figure 11.** Polygon with $\Theta(n^2)$ motorcycle intersections.

For each of the tested polygons without exception, the following interesting and encouraging fact could be observed: The number of structural changes, averaged over *all* insertion steps, is bounded by a small constant. The plots in Figures 12 and 13 give some details. We conclude that in practice our randomized insertion algorithm performs very well; $\mathcal{O}(1)$ structural changes per insertion leads to an expected overall running time of $\mathcal{O}(n \log^2 n)$.

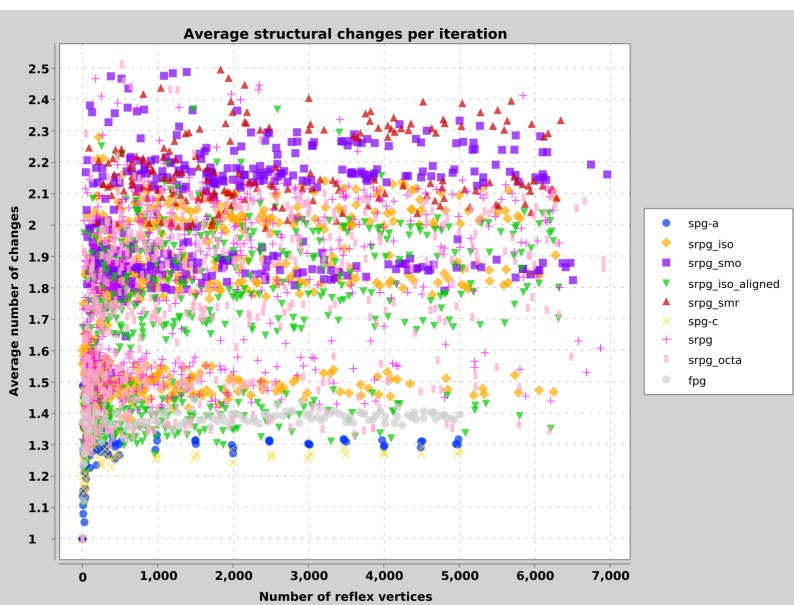

**Figure 12.** Average number of structural changes per insertion step. (For polygons from [10] with up to 7000 reflex vertices).

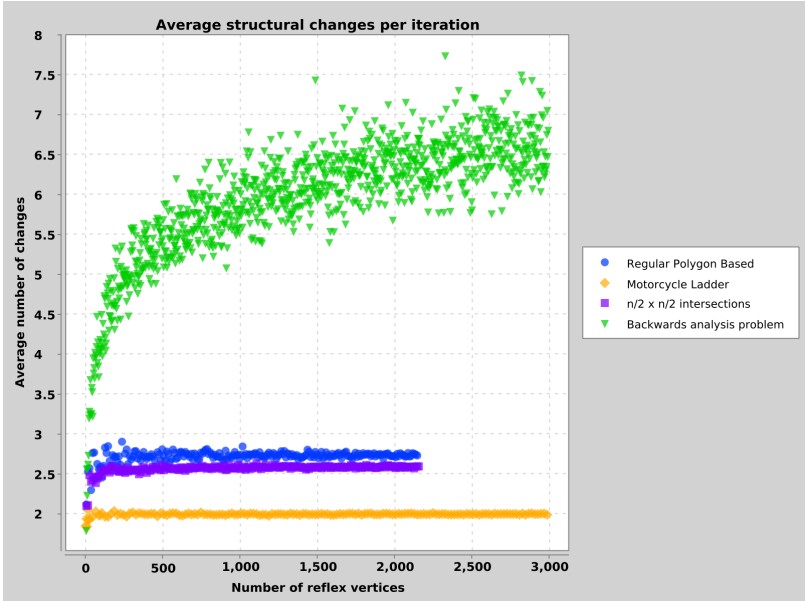

**Figure 13.** Average number of structural changes per insertion step, for specifically constructed polygons.

## 4. Structural Aspects

In this section we take a closer look at a structural property of polygons that has a direct impact on the performance of our insertion algorithm.

The most challenging configuration for the insertion algorithm is when all motorcycles of a polygon interact with each other, that is, when there are $\Theta(n^2)$ intersections among motorcycle traces to consider. However, for many types of polygons this will not be the case. The set of motorcycles then can be divided into subsets that can be processed independently by the insertion algorithm. If the sizes of these subsets are sufficiently small, much better theoretical running times can be achieved. In particular, for constant sizes a running time of $\mathcal{O}(n \log^2 n)$ is obtained.

### 4.1. Motorcycle Crews

Let us formalize the notion of interacting motorcycles. For this purpose we represent each motorcycle by a pair $(p, q)$ of its starting point and its potential crashing point on the polygon boundary $\partial \mathcal{P}$. Since we are only interested in whether these potential traces have intersection points within $\mathcal{P}$, we can simply model $\mathcal{P}$ by a disk and the traces by arcs within this disk. See Figure 14 for an example.

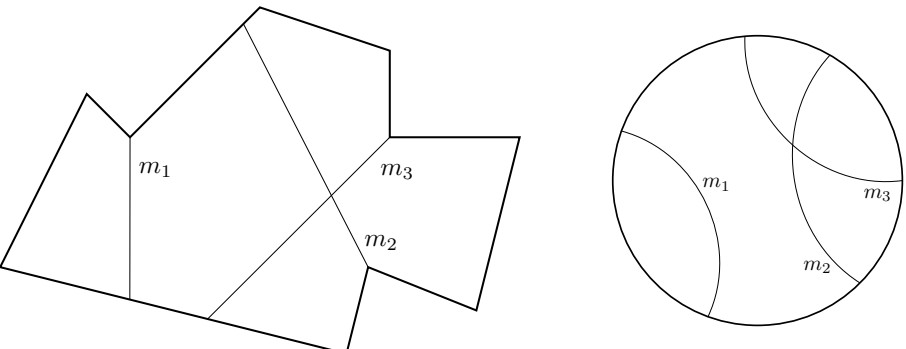

**Figure 14.** Polygon with motorcycles (**left**) and its abstraction on a disk (**right**).

In this way we obtain what is called the model of a *circle graph* (or *chord intersection graph*) in graph theory [14]. We are interested in the connected components of this graph, which we shall call *motorcycle crews* in our context. Each motorcycle crew can be processed independently by our insertion algorithm.

To compute these crews, we perform a counterclockwise walk along $\partial P$ from an arbitrary initial point. A list $\mathcal{L}$ is kept containing all currently encountered but not yet completed crews. When the starting point $p$ of a motorcycle trace $(p, q)$ is encountered, we add it to $\mathcal{L}$ and assign this motorcycle to a new crew. When the endpoint $q$ of a trace $(p, q)$ is encountered, we locate its starting point $p$ in $\mathcal{L}$ and merge all motorcycles between $p$ and $q$ into a common crew. If this crew is complete (meaning that each member has both its starting and its endpoint in $\mathcal{L}$), the crew is reported and removed from $\mathcal{L}$. Note that $\mathcal{L}$ can be implemented as a union-find data structure [15], such that all necessary operations can be carried out in $\mathcal{O}(n \log n)$ overall time.

### 4.2. Experimental Data

The algorithm for computing the motorcycle crews has been run on polygons with up to 30.000 vertices from the *Salzburg Database of Geometric Inputs* [10], as well as on polygons based on country outlines taken from *Natural Earth* [11] and *DIVA-GIS* [12].

Figure 15 shows the distribution of crew sizes in relation to the number of reflex vertices of a polygon. The smallest crew sizes strongly dominate all the others combined (a), which promises a speed-up of our insertion algorithm for most instances. A closer look at the remaining crew sizes (b) shows that there are a few hundred polygons (out of around 2700) that have one very large crew containing almost all motorcycles. For these polygons our insertion algorithm does not get a benefit for processing crews separately. For polygons based on country outlines the situation is less favorable; see Figure 16. Most polygons have crews of size $\mathcal{O}(n)$, that is, most motorcycles interact with each other.

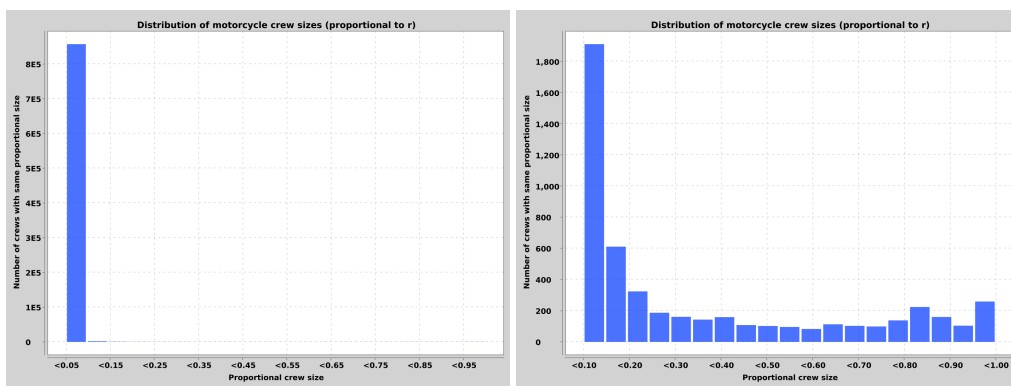

(**a**) All crew sizes.                  (**b**) Smallest crew sizes excluded.

**Figure 15.** Histogram of motorcycle crew sizes in relation to the number of reflex vertices. Data for 2700 polygons with at least 100 reflex vertices from *Salzburg Database of Geometric Inputs* [10].

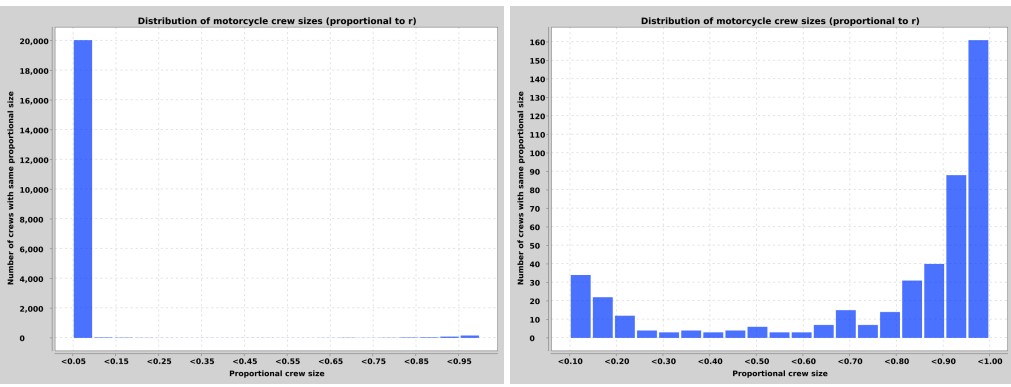

(**a**) All crew sizes.                  (**b**) Smallest crew sizes excluded.

**Figure 16.** Histogram of motorcycle crew sizes in relation to the number of reflex vertices. Data for 380 country outline based polygons with at least 100 reflex vertices from *Natural Earth* [11] and *DIVA-GIS* [12].

## 5. Conclusions

In summary, the insertion strategy enables a quick and simple construction of motorcycle graphs in practice. Together with the simple motorcycle-graph-based skeleton merging algorithm in [16], we obtain a new practical method for computing straight skeletons.

**Author Contributions:** Algorithm idea and initial design: M.S.; Algorithm refinement and article writing (including reviewing and editing): F.A. and M.S.; designing, implementing and executing tests on polygons: C.L. and M.S. All authors have read and agreed to the published version of the manuscript.

**Funding:** This research was supported by Project I 5270-N, Austrian Science Fund (FWF).

**Data Availability Statement:** Source code used in the practical part of this work can be found in the following Git repositories (all accessed on 6 May 2022): https://github.com/utnapischtim/motorcycle-graph, https://github.com/utnapischtim/sui, https://github.com/utnapischtim/qui, https://github.com/utnapischtim/otu. Analyzed data sets (polygons) are available at [10–12].

**Acknowledgments:** We thank the Computational Geometry and Applications Lab of the Department of Computer Science of the University of Salzburg for their database of geometric inputs [10].

**Conflicts of Interest:** The authors declare no conflict of interest.

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
