# Peer review of "Incremental Construction of Motorcycle Graphs"

_algorithms, doi:10.3390/a15070225_

Round 1

Reviewer 1 Report

There is a very large list of publications on the topic of the article. In my opinion, the problem considered in this paper is technically sufficiently difficult. The sequential processing algorithm of motorcycle graphs proposed by the authors looks quite natural. The comments about the computational complexity of the proposed algorithm are very useful. Illustration of the results on Figure 15 looks very convincing. So this paper deserves the opportunity to be published in Algorithms. There is a single may be optional wish to this paper. As for my part it is worthy to explain and discuss more detailed the randomization procedure in the presented algorithm.

Author Response

We have added explanations about the randomization procedure at the end of Section 2.3 as well as in the first half of Section 3.

Reviewer 2 Report

The present paper is an extended version of the conference paper published in the proceedings of the "38th European Workshop on Computational Geometry", Perugia, 2022.

The straight skeleton is a skeletal structure for polygons. The straight skeleton is determined by the results from moving inwards the polygon edges in a self-parallel way. The

polygon undergoes combinatorial and topological changes like edges shrinking to zero and, in

particular, the polygon might split several times. The vertices of the shrinking polygon trace

out a structure consisting of straight-line segments which is called the straight skeleton. The

re

ex vertices of the polygon move at individual speeds during the shrinking process and are

responsible for possible polygon splits. Their interaction is the motorcycle graph problem.

The union of the traces of all motorcycles constitutes the so-called motorcycle graph.

In this paper authors apply randomized insertion to the motorcycles and obtain an algorithm for computing the motorcycle graph which is simple and experimentally fast and

obtain a signi cant contribution to a new practical method for computing straight skeletons.

They also provide various test data which indicate its competitiveness to other implemented algorithms.

The paper is well written, with clearly given explanations and examples and also, clearly presented experimental results.

Therefore, I recommend the present paper to be published in Algorithms.

Author Response

We did some small changes as requested by Reviewer 1: to give more details on how randomization is used in our algorithm.
We have added respective explanations at the end of Section 2.3 as
well as in the first half of Section 3.
